# Expanding xylose metabolism in yeast for plant cell wall conversion to biofuels

Xin Li[1,2], Vivian Yaci Yu[1], Yuping Lin[1], Kulika Chomvong[3], Raíssa Estrela[1], Annsea Park[1], Julie M Liang[4], Elizabeth A Znameroski[1], Joanna Feehan[1], Soo Rin Kim[5,6], Yong-Su Jin[5,7], N Louise Glass[3], Jamie HD Cate[1,4,8]*

[1]Department of Molecular and Cell Biology, University of California, Berkeley, Berkeley, United States; [2]Impossible Foods, Inc, Redwood City, United States; [3]Department of Plant and Microbial Biology, University of California, Berkeley, Berkeley, United States; [4]Department of Chemistry, University of California, Berkeley, Berkeley, United States; [5]Institute for Genomic Biology, University of Illinois, Urbana, United States; [6]School of Food Science and Biotechnology, Kyungpook National University, Daegu, Republic of Korea; [7]Department of Food Science and Human Nutrition, University of Illinois, Urbana, United States; [8]Physical Biosciences Division, Lawrence Berkeley National Laboratory, Berkeley, United States

**Abstract** Sustainable biofuel production from renewable biomass will require the efficient and complete use of all abundant sugars in the plant cell wall. Using the cellulolytic fungus *Neurospora crassa* as a model, we identified a xylodextrin transport and consumption pathway required for its growth on hemicellulose. Reconstitution of this xylodextrin utilization pathway in *Saccharomyces cerevisiae* revealed that fungal xylose reductases act as xylodextrin reductases, producing xylosyl-xylitol oligomers as metabolic intermediates. These xylosyl-xylitol intermediates are generated by diverse fungi and bacteria, indicating that xylodextrin reduction is widespread in nature. Xylodextrins and xylosyl-xylitol oligomers are then hydrolyzed by two hydrolases to generate intracellular xylose and xylitol. Xylodextrin consumption using a xylodextrin transporter, xylodextrin reductases and tandem intracellular hydrolases in cofermentations with sucrose and glucose greatly expands the capacity of yeast to use plant cell wall-derived sugars and has the potential to increase the efficiency of both first-generation and next-generation biofuel production.

*For correspondence: jcate@lbl.gov

## Introduction

The biological production of biofuels and renewable chemicals from plant biomass requires an economic way to convert complex carbohydrate polymers from the plant cell wall into simple sugars that can be fermented by microbes (*Carroll and Somerville, 2009*; *Chundawat et al., 2011*). In current industrial methods, cellulose and hemicellulose, the two major polysaccharides found in the plant cell wall (*Somerville et al., 2004*), are generally processed into monomers of glucose and xylose, respectively (*Chundawat et al., 2011*). In addition to harsh pretreatment of biomass, large quantities of cellulase and hemicellulase enzyme cocktails are required to release monosaccharides from plant cell wall polymers, posing unsolved economic and logistical challenges (*Lynd et al., 2002*; *Himmel et al., 2007*; *Jarboe et al., 2010*; *Chundawat et al., 2011*). The bioethanol industry currently uses the yeast *Saccharomyces cerevisiae* to ferment sugars derived from cornstarch or sugarcane into ethanol (*Hong and Nielsen, 2012*), but *S. cerevisiae* requires substantial engineering to ferment sugars derived from plant cell walls such as cellobiose and xylose (*Kuyper et al., 2005*; *Jeffries, 2006*; *van Maris et al., 2007*; *Ha et al., 2011*; *Hong and Nielsen, 2012*; *Young et al., 2014*).

**eLife digest** Plants can be used to make 'biofuels', which are more sustainable alternatives to traditional fuels made from petroleum. Unfortunately, most biofuels are currently made from simple sugars or starch extracted from parts of plants that we also use for food, such as the grains of cereal crops.

Making biofuels from the parts of the plant that are not used for food—for example, the stems or leaves—would enable us to avoid a trade-off between food and fuel production. However, most of the sugars in these parts of the plant are locked away in the form of large, complex carbohydrates called cellulose and hemicellulose, which form the rigid cell wall surrounding each plant cell.

Currently, the industrial processes that can be used to make biofuels from plant cell walls are expensive and use a lot of energy. They involve heating or chemically treating the plant material to release the cellulose and hemicellulose. Then, large quantities of enzymes are added to break these carbohydrates down into simple sugars that can then be converted into alcohol (a biofuel) by yeast.

Fungi may be able to provide us with a better solution. Many species are able to grow on plants because they can break down cellulose and hemicellulose into simple sugars they can use for energy. If the genes involved in this process could be identified and inserted into yeast it may provide a new, cheaper method to make biofuels from plant cell walls.

To address this challenge, Li et al. studied how the fungus *Neurospora crassa* breaks down hemicellulose. This study identified a protein that can transport molecules of xylodextrin—which is found in hemicellulose—into the cells of the fungus, and two enzymes that break down the xylodextrin to make simple sugars, using a previously unknown chemical intermediate. When Li et al. inserted the genes that make the transport protein and the enzymes into yeast, the yeast were able to use plant cell wall material to make simple sugars and convert these to alcohol.

The yeast used more of the xylodextrin when they were grown with an additional source of energy, such as the sugars glucose or sucrose. Li et al.'s findings suggest that giving yeast the ability to break down hemicellulose has the potential to improve the efficiency of biofuel production. The next challenge will be to improve the process so that the yeast can convert the xylodextrin and simple sugars more rapidly.

## Results

In contrast to *S. cerevisiae*, many cellulolytic fungi including *Neurospora crassa* (*Tian et al., 2009*) naturally grow well on the cellulose and hemicellulose components of the plant cell wall. By using transcription profiling data (*Tian et al., 2009*) and by analyzing growth phenotypes of *N. crassa* knockout strains, we identified separate pathways used by *N. crassa* to consume cellodextrins (*Galazka et al., 2010*) and xylodextrins released by its secreted enzymes (*Figure 1A* and *Figure 1—figure supplement 1*). A strain carrying a deletion of a previously identified cellodextrin transporter (CDT-2, NCU08114) (*Galazka et al., 2010*) was unable to grow on xylan (*Figure 1—figure supplement 2*), and xylodextrins remained in the culture supernatant (*Figure 1—figure supplement 3*). As a direct test of transport function of CDT-2, *S. cerevisiae* strains expressing *cdt-2* were able to import xylobiose, xylotriose, and xylotetraose (*Figure 1—figure supplement 4*). Notably, *N. crassa* expresses a putative intracellular β-xylosidase, GH43-2 (NCU01900), when grown on xylan (*Sun et al., 2012*). Purified GH43-2 displayed robust hydrolase activity towards xylodextrins with a degree of polymerization (DP) spanning from 2 to 8, and with a pH optimum near 7 (*Figure 1—figure supplement 5*). The results with CDT-2 and GH43-2 confirm those obtained independently in *Cai et al. (2014)*. As with *cdt-1*, orthologues of *cdt-2* are widely distributed in the fungal kingdom (*Galazka et al., 2010*), suggesting that many fungi consume xylodextrins derived from plant cell walls. Furthermore, as with intracellular β-glucosidases (*Galazka et al., 2010*), intracellular β-xylosidases are also widespread in fungi (*Sun et al., 2012*) (*Figure 1—figure supplement 6*).

Cellodextrins and xylodextrins derived from plant cell walls are not catabolized by wild-type *S. cerevisiae* (*Matsushika et al., 2009*; *Galazka et al., 2010*; *Young et al., 2010*). Reconstitution of a cellodextrin transport and consumption pathway from *N. crassa* in *S. cerevisiae* enabled this yeast to ferment cellobiose (*Galazka et al., 2010*). We therefore reasoned that expression of a functional xylodextrin transport and consumption system from *N. crassa* might further expand the capabilities of

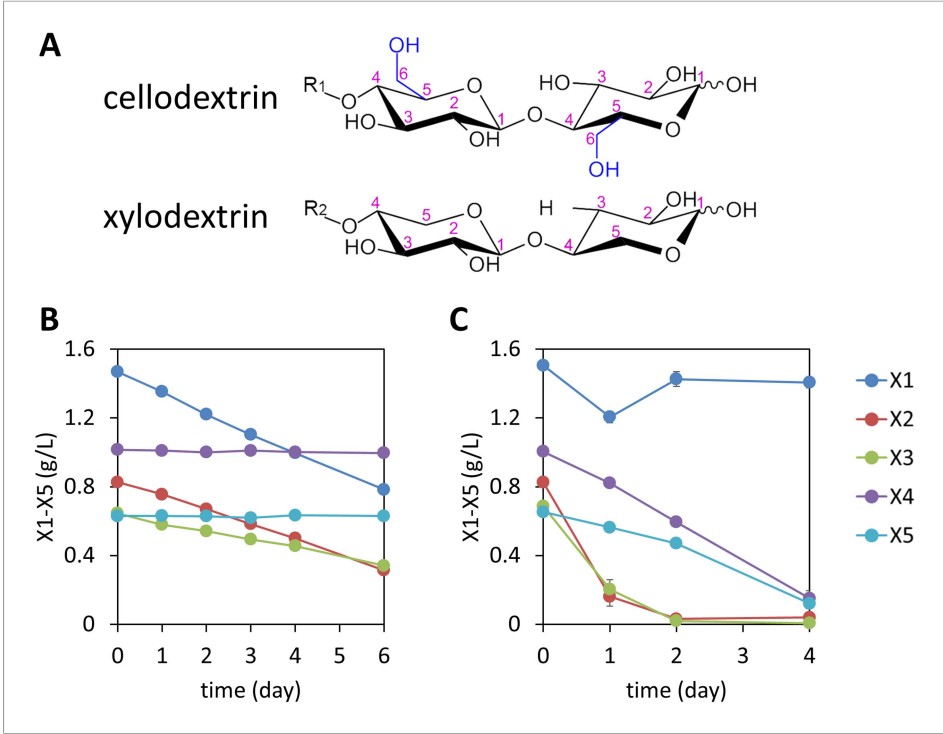

**Figure 1**. Consumption of xylodextrins by engineered *S. cerevisiae*. (**A**) Two oligosaccharide components derived from the plant cell wall. Cellodextrins, derived from cellulose, are a major source of glucose. Xylodextrins, derived from hemicellulose, are a major source of xylose. The 6-methoxy group (blue) distinguishes glucose derivatives from xylose. $R_1$, $R_2 = H$, cellobiose or xylobiose; $R_1 = β$-1,4-linked glucose monomers in cellodextrins of larger degrees of polymerization; $R_2 = β$-1,4-linked xylose monomers in xylodextrins of larger degrees of polymerization. (**B**) Xylose and xylodextrins remaining in a culture of *S. cerevisiae* grown on xylose and xylodextrins and expressing an XR/XDH xylose consumption pathway, CDT-2, and GH43-2, with a starting cell density of OD600 = 1 under aerobic conditions. (**C**) Xylose and xylodextrins in a culture as in (**B**) but with a starting cell density of OD600 = 20. In both panels, the concentrations of xylose (X1) and xylodextrins with higher DPs (X2–X5) remaining in the culture broth after different periods of time are shown. All experiments were conducted in biological triplicate, with error bars representing standard deviations.

The following figure supplements are available for figure 1:

**Figure supplement 1**. Transcriptional levels of transporters expressed in *N. crassa* grown on different carbon sources.

**Figure supplement 2**. Growth of *N. crassa* strains on different carbon sources.

**Figure supplement 3**. Xylodextrins in the xylan culture supernatant of the *N. crassa* Δcdt-2 strain.

**Figure supplement 4**. Transport of xylodextrins into the cytoplasm of *S. cerevisiae* strains expressing *N. crassa* transporters.

**Figure supplement 5**. Xylobiase activity of the predicted β-xylosidase GH43-2.

**Figure supplement 6**. Phylogenetic distribution of predicted intracellular β-xylosidases GH43-2 in filamentous fungi.

**Figure supplement 7**. Xylodextrin consumption profiles of *S. cerevisiae* strains lacking the xylodextrin pathway.

*S. cerevisiae* to utilize plant-derived xylodextrins. Previously, *S. cerevisiae* was engineered to consume xylose by introducing xylose isomerase (XI), or by introducing xylose reductase (XR) and xylitol dehydrogenase (XDH) (*Jeffries, 2006*; *van Maris et al., 2007*; *Matsushika et al., 2009*). To test

whether *S. cerevisiae* could utilize xylodextrins, a *S. cerevisiae* strain was engineered with the XR/XDH pathway derived from *Scheffersomyces stipitis*—similar to that in *N. crassa* (*Sun et al., 2012*)—and a xylodextrin transport (CDT-2) and consumption (GH43-2) pathway from *N. crassa*. The xylose utilizing yeast expressing CDT-2 along with the intracellular β-xylosidase GH43-2 was able to directly utilize xylodextrins with DPs of 2 or 3 (*Figure 1B* and *Figure 1—figure supplement 7*).

Notably, although high cell density cultures of the engineered yeast were capable of consuming xylodextrins with DPs up to 5, xylose levels remained high (*Figure 1C*), suggesting the existence of severe bottlenecks in the engineered yeast. These results mirror those of a previous attempt to engineer *S. cerevisiae* for xylodextrin consumption, in which xylose was reported to accumulate in the culture medium (*Fujii et al., 2011*). Analyses of the supernatants from cultures of the yeast strains expressing CDT-2, GH43-2 and the *S. stipitis* XR/XDH pathway surprisingly revealed that the xylodextrins were converted into xylosyl-xylitol oligomers, a set of previously unknown compounds rather than hydrolyzed to xylose and consumed (*Figure 2A* and *Figure 2—figure supplement 1*). The resulting xylosyl-xylitol oligomers were effectively dead-end products that could not be metabolized further.

Since the production of xylosyl-xylitol oligomers as intermediate metabolites has not been reported, the molecular components involved in their generation were examined. To test whether the xylosyl-xylitol oligomers resulted from side reactions of xylodextrins with endogenous *S. cerevisiae* enzymes, we used two separate yeast strains in a combined culture, one containing the xylodextrin hydrolysis pathway composed of CDT-2 and GH43-2, and the second with the XR/XDH xylose consumption pathway. The strain expressing CDT-2 and GH43-2 would cleave xylodextrins to xylose, which could then be secreted via endogenous transporters (*Hamacher et al., 2002*) and serve as a carbon source for the strain expressing the xylose consumption pathway (XR and XDH). The engineered yeast expressing XR and XDH is only capable of consuming xylose (*Figure 1B*). When co-cultured, these strains consumed xylodextrins without producing the xylosyl-xylitol byproduct (*Figure 2—figure supplement 2*). These results indicate that endogenous yeast enzymes and GH43-2 transglycolysis activity are not responsible for generating the xylosyl-xylitol byproducts, that is, that they must be generated by the XR from *S. stipitis* (*Ss*XR).

Fungal xylose reductases such as *Ss*XR have been widely used in industry for xylose fermentation. However, the structural details of substrate binding to the XR active site have not been established. To explore the molecular basis for XR reduction of oligomeric xylodextrins, the structure of *Candida tenuis* xylose reductase (*Ct*XR) (*Kavanagh et al., 2002*), a close homologue of *Ss*XR, was analyzed. *Ct*XR contains an open active site cavity where xylose could bind, located near the binding site for the NADH co-factor (*Kavanagh et al., 2002*; *Kratzer et al., 2006*). Notably, the open shape of the active site can readily accommodate the binding of longer xylodextrin substrates (*Figure 2B*). Using computational docking algorithms (*Trott and Olson, 2010*), xylobiose was found to fit well in the pocket. Furthermore, there are no obstructions in the protein that would prevent longer xylodextrin oligomers from binding (*Figure 2B*).

We reasoned that if the xylosyl-xylitol byproducts are generated by fungal XRs like that from *S. stipitis*, similar side products should be generated in *N. crassa*, thereby requiring an additional pathway for their consumption. Consistent with this hypothesis, xylose reductase XYR-1 (NCU08384) from *N. crassa* also generated xylosyl-xylitol products from xylodextrins (*Figure 2C*). However, when *N. crassa* was grown on xylan, no xylosyl-xylitol byproduct accumulated in the culture medium (*Figure 1—figure supplement 3*). Thus, *N. crassa* presumably expresses an additional enzymatic activity to consume xylosyl-xylitol oligomers. Consistent with this hypothesis, a second putative intracellular β-xylosidase upregulated when *N. crassa* was grown on xylan, GH43-7 (NCU09625) (*Sun et al., 2012*), had weak β-xylosidase activity but rapidly hydrolyzed xylosyl-xylitol into xylose and xylitol (*Figure 2D* and *Figure 2—figure supplement 3*). The newly identified xylosyl-xylitol-specific β-xylosidase GH43-7 is widely distributed in fungi and bacteria (*Figure 2E*), suggesting that it is used by a variety of microbes in the consumption of xylodextrins. Indeed, GH43-7 enzymes from the bacteria *Bacillus subtilis* and *Escherichia coli* cleave both xylodextrin and xylosyl-xylitol (*Figure 2F*).

To test whether xylosyl-xylitol is produced generally by microbes as an intermediary metabolite during their growth on hemicellulose, we extracted and analyzed the metabolites from a number of ascomycetes species and *B. subtilis* grown on xylodextrins. Notably, these widely divergent fungi and *B. subtilis* all produce xylosyl-xylitols when grown on xylodextrins (*Figure 3A* and *Figure 3—figure supplement 1*). These organisms span over 1 billion years of evolution (*Figure 3B*), indicating that the use of xylodextrin reductases to consume plant hemicellulose is widespread.

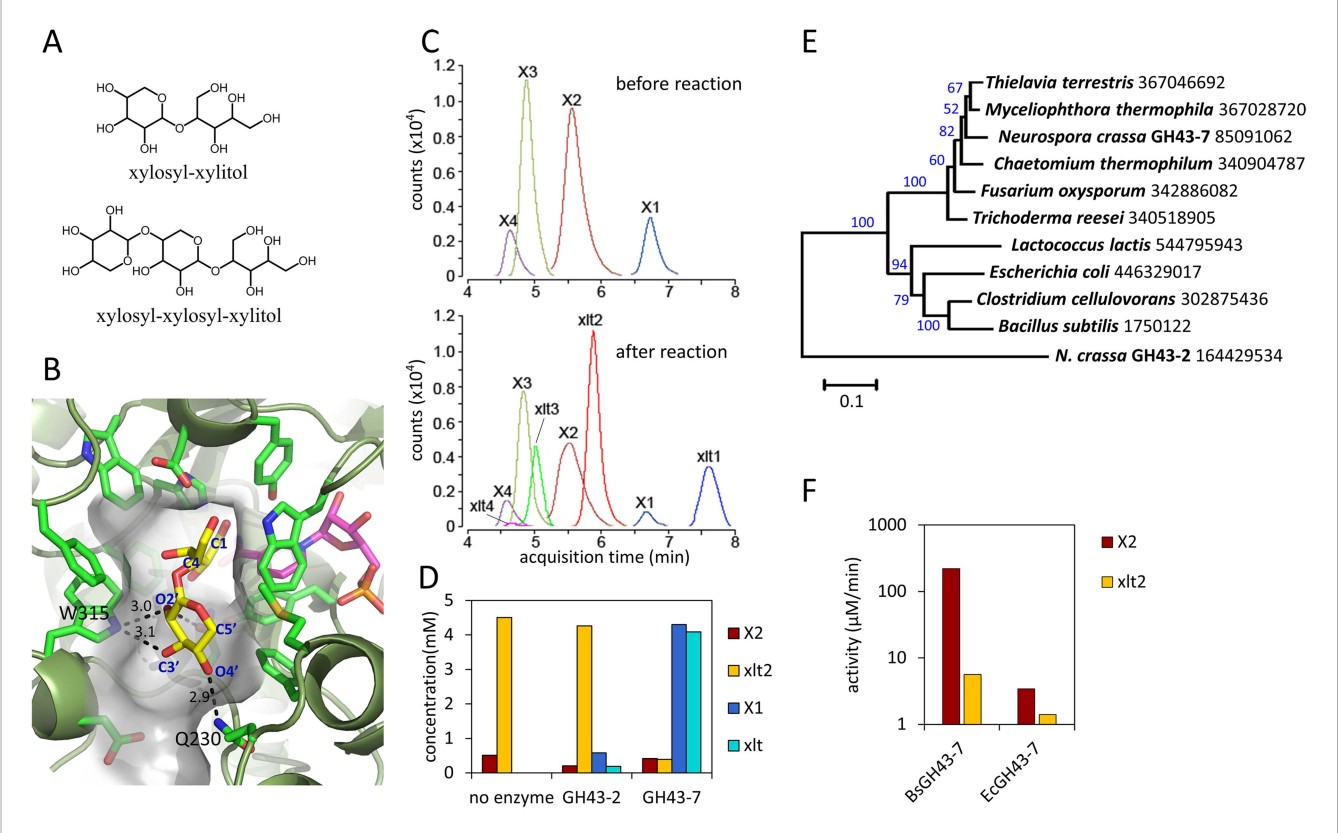

**Figure 2**. Production and enzymatic breakdown of xylosyl-xylitol. (**A**) Structures of xylosyl-xylitol and xylosyl-xylosyl-xylitol. (**B**) Computational docking model of xylobiose to *Ct*XR, with xylobiose in yellow, NADH cofactor in magenta, protein secondary structure in dark green, active site residues in bright green and showing side-chains. Part of the *Ct*XR surface is shown to depict the shape of the active site pocket. Black dotted lines show predicted hydrogen bonds between *Ct*XR and the non-reducing end residue of xylobiose. (**C**) Production of xylosyl-xylitol oligomers by *N. crassa* xylose reductase, XYR-1. Xylose, xylodextrins with DP of 2–4, and their reduced products are labeled X1–X4 and xlt1–xlt4, respectively. (**D**) Hydrolysis of xylosyl-xylitol by GH43-7. A mixture of 0.5 mM xylobiose and xylosyl-xylitol was used as substrates. Concentration of the products and the remaining substrates are shown after hydrolysis. (**E**) Phylogeny of GH43-7. *N. crassa* GH43-2 was used as an outgroup. 1000 bootstrap replicates were performed to calculate the supporting values shown on the branches. The scale bar indicates 0.1 substitutions per amino acid residue. The NCBI GI numbers of the sequences used to build the phylogenetic tree are indicated beside the species names. (**F**) Activity of two bacterial GH43-7 enzymes from *B. subtilis* (BsGH43-7) and *E. coli* (EcGH43-7).

The following figure supplements are available for figure 2:

**Figure supplement 1**. Xylosyl-xylitol oligomers generated in yeast cultures with xylodextrins as the sole carbon source.

**Figure supplement 2**. Xylodextrin metabolism by a co-culture of yeast strains to identify enzymatic source of xylosyl-xylitol.

**Figure supplement 3**. Chromatogram of xylosyl-xylitol hydrolysis products generated by **β**-xylosidases.

We next tested whether integration of the complete xylodextrin consumption pathway would overcome the poor xylodextrin utilization by *S. cerevisiae* (*Figure 1*) (*Fujii et al., 2011*). When combined with the original xylodextrin pathway (CDT-2 plus GH43-2), GH43-7 enabled *S. cerevisiae* to grow more rapidly on xylodextrin (*Figure 4A*) and eliminated accumulation of xylosyl-xylitol intermediates (*Figure 4B–D* and *Figure 4—figure supplement 1*). The presence of xylose and glucose greatly improved anaerobic fermentation of xylodextrins (*Figure 5* and *Figure 5—figure supplement 1* and *Figure 5—figure supplement 2*), indicating that metabolic sensing in *S. cerevisiae* with the complete xylodextrin pathway may require additional tuning (*Youk and van Oudenaarden, 2009*) for optimal xylodextrin fermentation. Notably, we observed

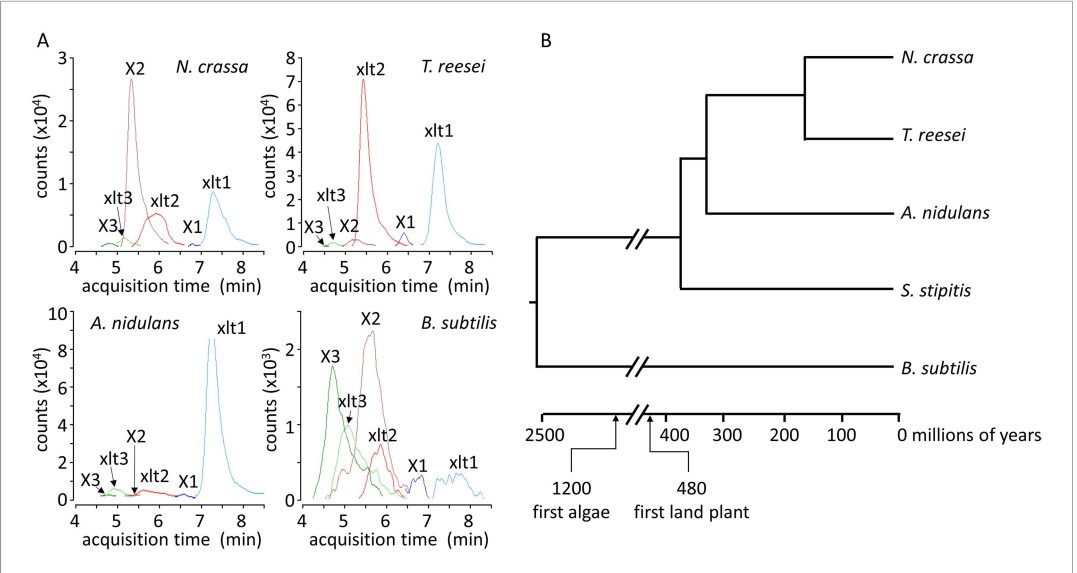

**Figure 3**. Xylosyl-xylitol and xylosyl-xylosyl-xylitol production by a range of microbes. (**A**) Xylodextrin-derived carbohydrate levels seen in chromatograms of intracellular metabolites for *N. crassa*, *T. reesei*, *A. nidulans* and *B. subtilis* grown on xylodextrins. Compounds are abbreviated as follows: X1, xylose; X2, xylobiose; X3, xylotriose; X4, xylotetraose; xlt, xylitol; xlt2, xylosyl-xylitol; xlt3, xylosyl-xylosyl-xylitol. (**B**) Phylogenetic tree of the organisms shown to produce xylosyl-xylitols during growth on xylodextrins. Ages taken from *Wellman et al. (2003)*; *Galagan et al. (2005)*; *Hedges et al. (2006)*.
The following figure supplement is available for figure 3:

**Figure supplement 1**. LC-MS/MS multiple reaction monitoring chromatograms of xylosyl-xylitols from cultures of microbes grown on xylodextrins.

that the XR/XDH pathway produced much less xylitol when xylodextrins were used in fermentations than from xylose (*Figure 5* and *Figure 5—figure supplement 2B*). Taken together, these results reveal that the XR/XDH pathway widely used in engineered *S. cerevisiae* naturally has broad substrate specificity for xylodextrins, and complete reconstitution of the naturally occurring xylodextrin pathway is necessary to enable *S. cerevisiae* to efficiently consume xylodextrins.

The observation that xylodextrin fermentation was stimulated by glucose (*Figure 5B*) suggested that the xylodextrin pathway could serve more generally for cofermentations to enhance biofuel production. We therefore tested whether xylodextrin fermentation could be carried out simultaneously with sucrose fermentation, as a means to augment ethanol yield from sugarcane. In this scenario, xylodextrins released by hot water treatment (*Hendriks and Zeeman, 2009*; *Agbor et al., 2011*; *Vallejos et al., 2012*) could be added to sucrose fermentations using yeast engineered with the xylodextrin consumption pathway. To test this idea, we used strain SR8U engineered with the xylodextrin pathway (CDT-2, GH43-2, and GH43-7) in fermentations combining sucrose and xylodextrins. We observe simultaneous fermentation of sucrose and xylodextrins, with increased ethanol yields (*Figure 6*). Notably, the levels of xylitol production were found to be low (*Figure 6*), as observed in cofermentations with glucose (*Figure 5B*).

## Discussion

Using yeast as a test platform, we identified a xylodextrin consumption pathway in *N. crassa* (*Figure 7*) that surprisingly involves a new metabolic intermediate widely produced in nature by many fungi and bacteria. In bacteria such as *B. subtilis*, xylosyl-xylitol may be generated by aldo-keto reductases known to possess broad substrate specificity (*Barski et al., 2008*). The discovery of the xylodextrin

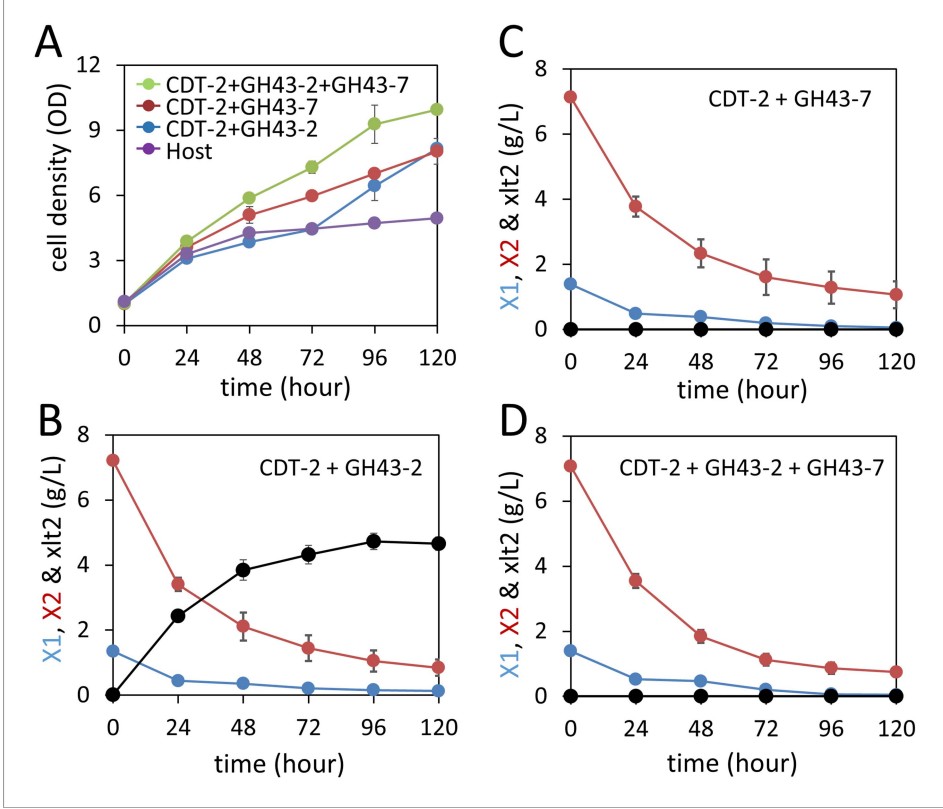

**Figure 4**. Aerobic consumption of xylodextrins with the complete xylodextrin pathway. (**A**) Yeast growth curves with xylodextrin as the sole carbon source under aerobic conditions with a cell density at OD600 = 1. Yeast strain SR8U without plasmids, or transformed with plasmid expressing CDT-2 and GH43-2 (pXD8.4), CDT-2 and GH43-7 (pXD8.6) or all three genes (pXD8.7) are shown. (**B–D**) Xylobiose consumption with xylodextrin as the sole carbon source under aerobic conditions with a cell density of OD600 = 20. Xylosyl-xylitol (xlt2) accumulation was only observed in the SR8U strain bearing plasmid pXD8.4, that is, lacking GH43-7. Error bars represent standard deviations of biological triplicates (panels **A–D**).

The following figure supplement is available for figure 4:

**Figure supplement 1**. Culture media composition during yeast growth on xylodextrin.

consumption pathway along with cellodextrin consumption (*Galazka et al., 2010*) in cellulolytic fungi for the two major sugar components of the plant cell wall now provides many modes of engineering yeast to ferment plant biomass-derived sugars (*Figure 7*). An alternative xylose consumption pathway using xylose isomerase could also be used with the xylodextrin transporter and xylodextrin hydrolase GH43-2 (*van Maris et al., 2007*). However, the XR/XDH pathway may provide significant advantages in realistic fermentation conditions with sugars derived from hemicellulose. The breakdown of hemicellulose, which is acetylated (*Sun et al., 2012*), releases highly toxic acetate, degrading the performance of *S. cerevisiae* fermentations (*Bellissimi et al., 2009*; *Sun et al., 2012*). The cofactor imbalance problem of the XR/XDH pathway, which can lead to accumulation of reduced byproducts (xylitol and glycerol) and therefore was deemed a problem, can be exploited to drive acetate reduction, thereby detoxifying the fermentation medium and increasing ethanol production (*Wei et al., 2013*).

With optimization, that is, through improvements to xylodextrin transporter performance and chromosomal integration (*Ryan et al., 2014*), the newly identified xylodextrin consumption pathway provides new opportunities to expand first-generation bioethanol production from cornstarch or sugarcane to include hemicellulose from the plant cell wall. For example, we propose that xylodextrins released from the hemicellulose in sugarcane bagasse by using compressed hot water treatment (*Hendriks and Zeeman, 2009*; *Agbor et al., 2011*; *Vallejos et al., 2012*) could be directly fermented

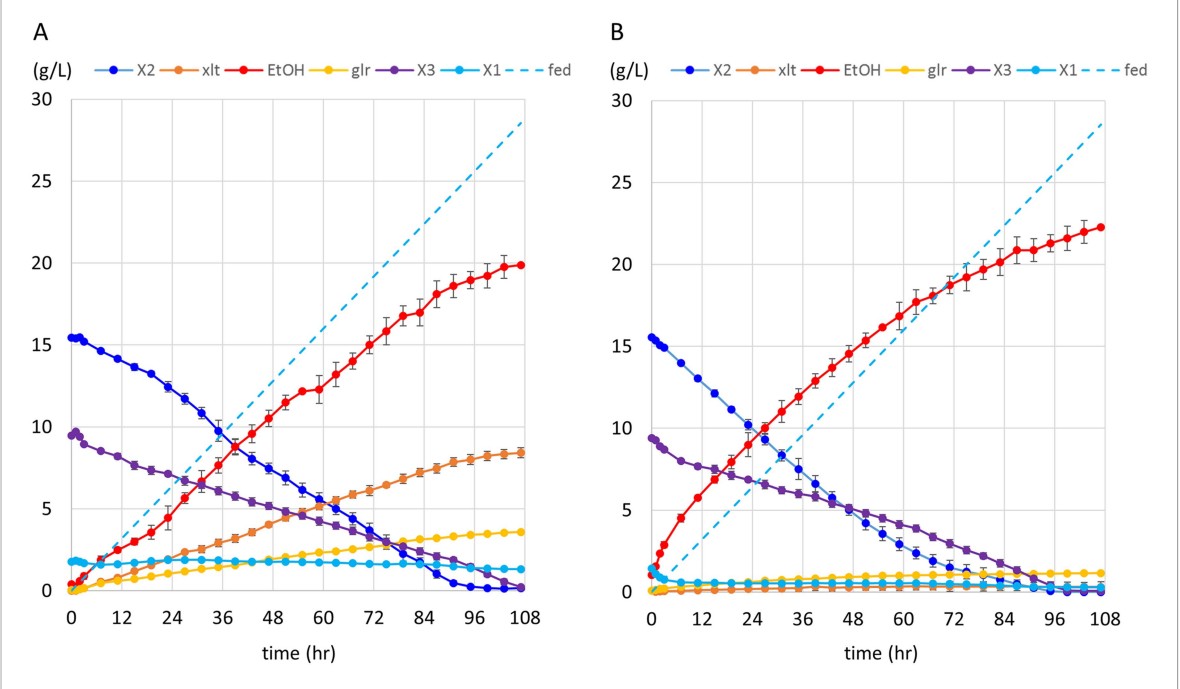

**Figure 5**. Anaerobic fermentation of xylodextrins in co-fermentations with xylose or glucose. (**A**) Anaerobic fermentation of xylodextrins and xylose, in a fed-batch reactor. Strain SR8U expressing CDT-2, GH43-2, and GH43-7 (plasmid pXD8.7) was used at an initial OD600 of 20. Solid lines represent concentrations of compounds in the media. Blue dotted line shows the total amount of xylose added to the culture over time. Error bars represent standard deviations of biological duplicates. (**B**) Anaerobic fermentation of xylodextrins and glucose, in a fed-batch reactor. Glucose was not detected in the fermentation broth. Error bars represent standard deviations of biological duplicates.

The following figure supplements are available for figure 5:

**Figure supplement 1**. Anaerobic xylodextrin utilization in the presence of xylose.

**Figure supplement 2**. Control anaerobic fermentations with *S. cerevisiae* strain expressing the complete xylodextrin utilization pathway.

by yeast engineered to consume xylodextrins, as we have shown in proof-of-principle experiments (*Figure 6*). Xylodextrin consumption combined with glucose or cellodextrin consumption (*Figure 7*) could also improve next-generation biofuel production from lignocellulosic feedstocks under a number of pretreatment scenarios (*Hendriks and Zeeman, 2009*; *Vallejos et al., 2012*). These pathways could find widespread use to overcome remaining bottlenecks to fermentation of lignocellulosic feedstocks as a sustainable and economical source of biofuels and renewable chemicals.

## Materials and methods

### *Neurospora crassa* strains

*N. crassa* strains obtained from the Fungal Genetics Stock Center (FGSC) (*McCluskey et al., 2010*) include the WT (FGSC 2489), and deletion strains for the two oligosaccharide transporters: NCU00801 (FGSC 16575) and NCU08114 (FGSC 17868) (*Colot et al., 2006*).

### *Neurospora crassa* growth assays

Conidia were inoculated at a concentration equal to $10^6$ conidia per ml in 3 ml Vogel's media (*Vogel, 1956*) with 2% wt/vol powdered *Miscanthus giganteus* (Energy Bioscience Institute, UC-Berkeley), Avicel PH 101 (Sigma-Aldrich, St. Louis, MO), beechwood xylan (Sigma-Aldrich), or pectin (Sigma-Aldrich) in a 24-well deep-well plate. The plate was sealed with Corning breathable sealing tape and

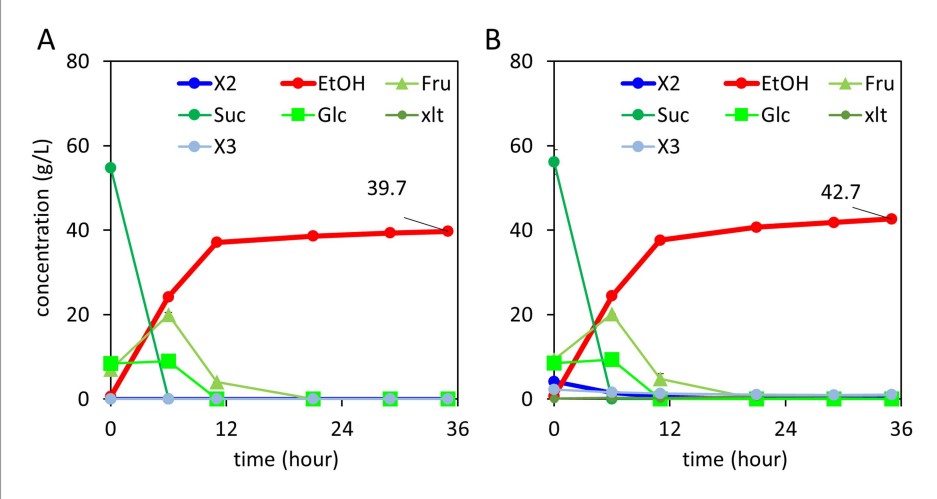

**Figure 6**. Xylodextrin and sucrose co-fermentations. (**A**) Sucrose fermentation. Vertical axis, g/l; horizontal axis, time in hours. (**B**) Xylodextrin and sucrose batch co-fermentation using strain SR8U expressing CDT-2, GH43-2, and GH43-7 (plasmid pXD8.7). Vertical axis, g/l; horizontal axis, time in hours. The xylodextrins were supplied at 10 g/l which containing xylobiose (4.2 g/l) and xylotriose (2.3 g/l). Not fermented in the timeframe of this experiment, the xylodextrin sample also included xylotetraose and xylopentaose, in addition to hemicellulose modifiers such as acetate.

incubated at 25°C in constant light and with shaking (200 rpm). Images were taken at 48 hr. Culture supernatants were diluted 200 times with 0.1 M NaOH before Dionex high-performance anion exchange chromatographic (HPAEC) analysis, as described below. *N. crassa* growth on xylan was also determined by measuring *N. crassa* biomass accumulation. *N. crassa* grown on xylan for 3 days was harvested by filtration over a Whatman glass microfiber filter (GF/F) on a Büchner funnel and washed with 50 ml water. Biomass was then collected from the filter, dried in a 70°C oven, and weighed.

## Plasmids and yeast strains

Template gDNA from the *N. crassa* WT strain (FGSC 2489) and from the *S. cerevisiae* S288C strain was extracted as described in http://www.fgsc.net/fgn35/lee35.pdf (*McCluskey et al., 2010*). Open reading frames (ORFs) of the β-xylosidase genes NCU01900 and NCU09652 (GH43-2 and GH43-7) were amplified from the *N. crassa* gDNA template. For biochemical assays, each ORF was fused with a C-terminal His$_6$-tag and flanked with the *S. cerevisiae* P$_{TEF1}$ promoter and *CYC1* transcriptional terminator in the 2μ yeast plasmid pRS423 backbone. Plasmid pRS426_NCU08114 was described previously (*Galazka et al., 2010*). Plasmid pLNL78 containing the xylose utilization pathway (xylose reductase, xylitol dehydrogenase, and xylulose kinase) from *S. stipitis* was obtained from the lab of John Dueber (*Latimer et al., 2014*). Plasmid pXD2, a single-plasmid form of the xylodextrin pathway, was constructed by integrating NCU08114 (CDT-2) and

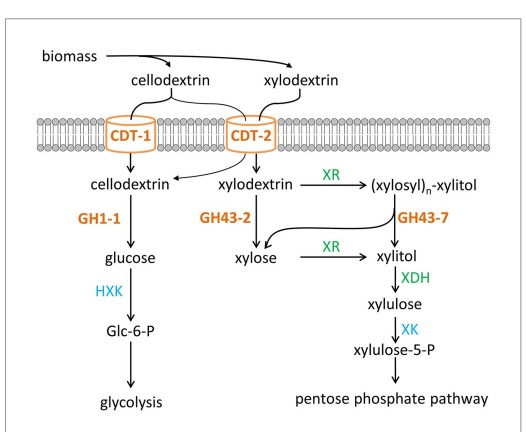

**Figure 7**. Two pathways of oligosaccharide consumption in *N. crassa* reconstituted in *S. cerevisiae*. Intracellular cellobiose utilization requires CDT-1 or CDT-2 along with β-glucosidase GH1-1 (*Galazka et al., 2010*) and enters glycolysis after phosphorylation by hexokinases (HXK) to form glucose-6-phosphate (Glc-6-P). Intracellular xylodextrin utilization also uses CDT-2 and requires the intracellular β-xylosidases GH43-2 and GH43-7. The resulting xylose can be assimilated through the pentose phosphate pathway consisting of xylose/xylodextrin reductase (XR), xylitol dehydrogenase (XDH), and xylulokinase (XK).

NCU01900 (GH43-2) expression cassettes into pLNL78, using the In-Fusion Cloning Kit (Clontech). Plasmid pXD8.4 derived from plasmid pRS316 (*Sikorski and Hieter, 1989*) was used to express CDT-2 and GH43-2, each from the $P_{CCW12}$ promoter. Plasmid pXD8.6 was derived from pXD8.4 by replacing the GH43-2 ORF with the ORF for GH43-7. pXD8.7 contained all three expression cassettes (CDT-2, GH43-2, and GH43-7) using the $P_{CCW12}$ promoter for each. *S. cerevisiae* strain D452-2 (*MAT*a *leu2 his3 ura3 can1*) (*Kurtzman, 1994*) and SR8U (the uracil autotrophic version of the evolved xylose fast utilization strain SR8) (*Kim et al., 2013*) were used as recipient strains for the yeast experiments. The ORF for *N. crassa* xylose reductase (*xyr-1*, NcXR) was amplified from *N. crassa* gDNA and the introns were removed by overlapping PCR. XR ORF was fused to a C-terminal $His_6$-tag and flanked with the *S. cerevisiae* $P_{CCW12}$ promoter and *CYC1* transcriptional terminator and inserted into plasmid pRS313.

A list of the plasmids used in this study can be found in *Table 1*.

## Yeast cell-based xylodextrin uptake assay

*S. cerevisiae* was grown in an optimized minimum medium (oMM) lacking uracil into late log phase. The oMM contained 1.7 g/l YNB (Sigma-Aldrich, Y1251), twofold appropriate CSM dropout mixture, 10 g/l $(NH_4)_2SO_4$, 1 g/l $MgSO_4.7H_2O$, 6 g/l $KH_2PO_4$, 100 mg/l adenine hemisulfate, 10 mg/l inositol, 100 mg/l glutamic acid, 20 mg/l lysine, 375 mg/l serine, and 100 mM 4-morpholineethanesulfonic acid (MES), pH 6.0 (*Lin et al., 2014*). Cells were then harvested and washed three times with assay buffer (5 mM MES, 100 mM NaCl, pH 6.0) and resuspended to a final OD600 of 40. Substrate stocks were prepared in the same assay buffer at a concentration of 200 µM. Transport assays were initiated by mixing equal volumes of the cell suspension and the substrate stock. Reactions were incubated at 30°C with continuous shaking for 30 min. Samples were centrifuged at 14,000 rpm at 4°C for 5 min to remove yeast cells. 400 µl of each sample supernatant was transferred to an HPLC vial containing 100 µl 0.5 M NaOH, and the concentration of the remaining substrate was measured by HPAEC as described below.

## Enzyme purification

*S. cerevisiae* strains transformed with pRS423_GH43-2, pRS423_GH43-7, or pRS313_NcXR were grown in oMM lacking histidine with 2% glucose until late log phase before harvesting by centrifugation. *E. coli* strains BL21DE3 transformed with pET302_BsGH43-7 or pET302_EcGH43-7 were grown in TB medium, induced with 0.2 mM IPTG at OD600 of 0.8, and harvested by centrifugation 12 hr after induction. Yeast or *E. coli* cell pellets were resuspended in a buffer containing 50 mM Tris–HCl, 100 mM NaCl, 0.5 mM DTT, pH 7.4 and protease inhibitor cocktail (Pierce Biotechnology, Rockford, IL). Cells were lysed with an Avestin homogenizer, and the clarified supernatant was loaded onto a HisTrap column (GE Healthcare, Sweden). His-tagged enzymes were

**Table 1**. A list of plasmids used in this study

| Plasmid | Genotype and use | Use | Ref. |
|---|---|---|---|
| pRS426_NCU08114 | $P_{PGK1}$-CDT-2 | transport assay | (*Galazka et al., 2010*) |
| pRS423_GH43-2 | $P_{TEF1}$-GH43-2 | enzyme purification | this study |
| pRS423_GH43-7 | $P_{TEF1}$-GH43-7 | enzyme purification | this study |
| pRS313_NcXR | $P_{CCW12}$-NcXR | enzyme purification | this study |
| pET302_EcGH43-7 | EcGH43-7 | enzyme purification | this study |
| pET302_BsGH43-7 | BsGH43-7 | enzyme purification | this study |
| pLNL78 | $P_{RNR2}$-SsXK::$P_{TEF1}$-SsXR::$P_{TEF1}$-SsXDH | fermentation | (*Galazka et al., 2010*) |
| pXD2 | $P_{RNR2}$-SsXK::$P_{TEF1}$-SsXR::$P_{TEF1}$-SsXDH::$P_{PGK1}$-CDT-2::$P_{TEF1}$-GH43-2 | fermentation | this study |
| pXD8.4 | $P_{CCW12}$-CDT-2::$P_{CCW12}$-GH43-2 | fermentation | this study |
| pXD8.6 | $P_{CCW12}$-CDT-2::$P_{CCW12}$-GH43-7 | fermentation | this study |
| pXD8.7 | $P_{CCW12}$-CDT-2::$P_{CCW12}$-GH43-7::$P_{CCW12}$-GH43-7 | fermentation | this study |

purified with an imidazole gradient, buffer-exchanged into 20 mM Tris–HCl, 100 mM NaCl, pH 7.4, and concentrated to 5 mg/ml.

## Enzyme assays

For the β-xylosidase assay of GH43-2 with xylodextrins, 0.5 μM of purified enzyme was incubated with 0.1% in-house prepared xylodextrin or 1 mM xylobiose (Megazyme, Ireland) in 1× PBS at 30°C. Reactions were sampled at 30 min and quenched by adding 5 vol of 0.1 M NaOH. The products were analyzed by HPAEC as described below. For pH profiling, acetate buffer at pH 4.0, 4.5, 5.0, 5.5, 6.0, and phosphate buffer at 6.5, 7.0, 7.5, 8 were added at a concentration of 0.1 M. For the β-xylosidase assay of GH43-2 and GH43-7 with xylosyl-xylitol, 10 μM of purified enzyme was incubated with 4.5 mM xylosyl-xylitol and 0.5 mM xylobiose in 20 mM MES buffer, pH = 7.0, and 1 mM $CaCl_2$ at 30°C. Reactions were sampled at 3 hr and quenched by heating at 99°C for 10 min. The products were analyzed by ion-exclusion HPLC as described below.

For the xylose reductase assays of NcXR, 1 μM of purified enzyme was incubated with 0.06% xylodextrin and 2 mM NADPH in 1× PBS at 30°C. Reactions were sampled at 30 min and quenched by heating at 99°C for 10 min. The products were analyzed by LC-QToF as described below.

## Oligosaccharide preparation

Xylodextrin was purchased from Cascade Analytical Reagents and Biochemicals or prepared according to published procedures (*Akpinar et al., 2009*) with slight modifications. In brief, 20 g beechwood xylan (Sigma–Aldrich) was fully suspended in 1000 ml water, to which 13.6 ml 18.4 M $H_2SO_4$ was added. The mixture was incubated in a 150°C oil bath with continuous stirring. After 30 min, the reaction was poured into a 2-L plastic container on ice, with stirring to allow it to cool. Then 0.25 mol $CaCO_3$ was slowly added to neutralize the pH and precipitate sulfate. The supernatant was filtered and concentrated on a rotary evaporator at 50°C to dryness. The in-house prepared xylodextrin contained about 30% xylose monomers and 70% oligomers. To obtain a larger fraction of short chain xylodextrin, the commercial xylodextrin was dissolved to 20% wt/vol and incubated with 2 mg/ml xylanase at 37°C for 48 hr. Heat deactivation and filtration were performed before use.

Xylosyl-xylitol was purified from the culture broth of strain SR8-containing plasmids pXD8.4 in xylodextrin medium. 50 ml of culture supernatant was concentrated on a rotary evaporator at 50°C to about 5 ml. The filtered sample was loaded on an XK 16/70 column (GE Healthcare) packed with Supelclean ENVI-Carb (Sigma–Aldrich) mounted on an ÄKTA Purifier (GE Healthcare). The column was eluted with a gradient of acetonitrile at a flow rate of 3.0 ml/min at room temperature. Purified fractions, verified by LC-MS, were pooled and concentrated. The final product, containing 90% of xylosyl-xylitol and 10% xylobiose, was used as the substrate for enzyme assays and as an HPLC calibration standard.

## Measurement of xylosyl-xylitol production by fungi and *B. subtilis*

*N. crassa* strain (FGSC 2489) and *Aspergillus nidulans* were stored and conidiated on agar slants of Volgel's medium (*Vogel, 1956*) with 2% glucose. *Trichoderma reesei* (strain QM6a) was conidiated on potato dextrose agar (PDA) plates. Condia from each fungi were collected by resuspending in water and used for inoculation at a concentration of $10^6$ cells per ml. *N. crassa* and *A. nidulans* were inoculated into Volgel's medium with 2% xylodextrin. *T. reesei* was inoculated into *Trichoderma* minimal medium (*Penttilä et al., 1987*) with 2% xylodextrin. *N. crassa*, *A. nidulans*, and *T. reesei* were grown in shaking flasks at 25°C, 37°C, and 30°C respectively. After 40 hr, mycelia from 2 ml of culture were harvested and washed with water on a glass fiber filter and transferred to a pre-chilled screw-capped 2 ml tube containing 0.5 ml Zirconia beads (0.5 mm) and 1.2 ml acidic acetonitrile extraction solution (80% Acetonitrile, 20% $H_2O$, and 0.1 M formic acid, [*Rabinowitz and Kimball, 2007*]). The tubes were then plunged into liquid nitrogen. The harvest process was controlled within 30 s. Samples were kept at −80°C until extraction, as described below.

*B. sublitis* was stored on 0.5× LB (1% tryptone, 0.5% yeast extract, and 0.5% NaCl) agar plates. A single colony was inoculated into 0.5× LB liquid medium with 1% glucose and allowed to grow in a 37°C shaker overnight. An inoculum from the overnight culture was transferred to fresh 0.5× LB liquid medium with 1% xylodextrin at an initial $OD_{600}$ of 0.2. After 40 hr, 2 ml of the culture was spun down and washed with cold PBS solution. Zirconia beads and acidic acetonitrile extraction solution

were added to the cell pellet. The tubes were then flash frozen immediately and kept at −80°C until extraction.

For extraction, all samples were allowed to thaw at 4°C for 10 min, bead beat for 2 min, and vortexed at 4°C for 20 min. 50 µl of the supernatant from each sample was analyzed by LC-MS/MS (see 'Mass spectrometric analyses' section).

### Aerobic yeast cultures with xylodextrins

Yeast strains were pre-grown aerobically overnight in oMM medium containing 2% glucose, washed three times with water, and resuspended in oMM medium. For aerobic growth, strains were inoculated at a starting OD600 of 1.0 or 20 in 50 ml oMM medium with 3% wt/vol xylodextrins and cultivated in 250 ml Erlenmeyer flasks covered with four layers of miracle cloth, shaking at 220 rpm. At the indicated time points, 0.8 ml samples were removed and pelleted. 20 µl supernatants were analyzed by ion-exclusion HPLC to determine xylose, xylitol, glycerol, and ethanol concentrations. 25 µl of 1:200 diluted or 2 µl of 1:100 diluted supernatant was analyzed by HPAEC or LC-QToF, respectively, to determine xylodextrin concentrations.

### Fed-batch anaerobic fermentations

Anaerobic fermentation experiments were performed in a 1-L stirred tank bioreactor (DASGIP Bioreactor system, Type DGCS4, Eppendorf AG, Germany), containing oMM medium with 3% wt/vol xylodextrins inoculated with an initial cell concentration of OD600 = 20. The runs were performed at 30°C for 107 hr. The culture was agitated at 200 rpm and purged constantly with 6 l/hr of nitrogen. For xylose plus xylodextrin co-fermentations, xylose was fed continuously at 0.8 ml/hr from a 25% stock. During the fermentation, 3 ml cell-free samples were taken each 4 hr with an autosampler through a ceramic sampling probe (Seg-Flow Sampling System, Flownamics, Madison, WI). 20 µl of the supernatant fraction were analyzed by ion-exclusion HPLC to determine xylose, xylitol, glycerol, acetate, and ethanol concentrations. 2 µl of 1:100 diluted supernatant was analyzed by LC-QToF to determine xylodextrin concentrations. For glucose plus xylodextrin co-fermentations, glucose was fed continuously at 2 ml/hr from a 10% stock. Analytes were detected as described for xylose plus xylodextrin co-fermentations, with the addition of the measurement of glucose concentrations in the culture broth.

### Co-fermentation of sucrose plus xylodextrins

Yeast strain SR8U with plasmid pXD8.7 was pre-grown aerobically to late-log phase in oMM medium lacking uracil and containing 2% glucose, washed with water, and resuspended in oMM medium. Media containing 75 g/l sucrose plus or minus 15 g/l xylodextrins were inoculated with 20 OD of the washed yeast seed culture and purged with $N_2$. Fermentations were carried out in 50 ml of oMM medium in 125 ml serum bottles shaking at 220 rpm in a 30°C shaker. At the indicated time points, 1 ml samples were removed and pelleted. 5 µl supernatants were analyzed by ion-exclusion HPLC to determine sucrose, glucose, fructose, xylose, xylitol, glycerol, and ethanol concentrations. 2 µl of 1:100 diluted supernatant was analyzed by LC-QToF, as described below, to determine xylodextrin concentrations.

### Ion-exclusion HPLC analysis

Ion-exclusion HPLC was performed on a Prominence HPLC (Shimadzu, Japan) equipped with a refractive index detector. Xylose fermentation samples were resolved on a Rezex RFQ-Fast Fruit H+ 8% column (100 × 7.8 mm, Phenomenex, Torrance, CA) using a flow rate of 1 ml/min at 50°C. Xylodextrin fermentation samples were resolved on Aminex HPX-87H Column (300 × 7.8 mm, Bio-Rad, Hercules, CA) at a flow rate of 0.6 ml/min at 40°C. Both columns used a mobile phase of 0.01 N $H_2SO_4$.

### HPAEC analysis

HPAEC analysis was performed on a ICS-3000 HPLC (Thermo Fisher, Sunnyvale, CA) using a CarboPac PA200 analytical column (150 × 3 mm) and a CarboPac PA200 guard column (3 × 30 mm) at 30°C. Following injection of 25 µl of diluted samples, elution was performed at 0.4 ml/min using 0.1 M NaOH in the mobile phase with sodium acetate gradients. For xylodextrin and xylosyl-xylitol separation, the acetate gradients were 0 mM for 1 min, increasing to 80 mM in 8 min, increasing to

300 mM in 1 min, keeping at 30 mM for 2 min, followed by re-equilibration at 0 mM for 3 min. Carbohydrates were detected using pulsed amperometric detection (PAD) and peaks were analyzed and quantified using the Chromeleon software package.

## Mass spectrometric analyses

All mass spectrometric analyses were performed on an Agilent 6520 Accurate-Mass Q-TOF coupled with an Agilent 1200 LC (Agilent Technologies, Santa Clara, CA). Samples were resolved on a 100 × 7.8 mm Rezex RFQ-Fast Fruit H+ 8% column (Phenomenex) using a mobile phase of 0.5% formic acid at a flow rate of 0.3 ml/min at 55˚C.

To determine the accurate masses of the unknown metabolites, 2 µl of 1:100 diluted yeast culture supernatant was analyzed by LC-QToF. Nitrogen was used as the instrument gas. The source voltage (Vcap) was 3000 V in negative ion mode, and the fragmentor was set to 100 V. The drying gas temperature was 300˚C; drying gas flow was 7 l/min; and nebulizer pressure was 45 psi. The ESI source used a separate nebulizer for the continuous, low-level introduction of reference mass compounds (112.985587, 1033.988109) to maintain mass axis calibration. Data were collected at an acquisition rate of 1 Hz from m/z 50 to 1100 and stored in centroid mode.

LC-MS/MS was performed to confirm the identity of xylosyl-xylitol and xylosyl-xylosyl-xylitol. The compound with a retention time (RT) of 5.8 min and m/z ratio of 283.103 and the compound with an RT of 4.7 min and m/z ratio of 415.15 were fragmented with collision energies of 10, 20, and 40 eV. MS/MS spectra were acquired, and the product ions were compared and matched to the calculated fragment ions generated by the Fragmentation Tools in ChemBioDraw Ultra v13.

To quantify the carbohydrates and carbohydrate derivatives in the culture, culture supernatants were diluted 100-fold in water and 2 µl was analyzed by LC-QToF. Spectra were imported to Qualtitative Analysis module of Agilent MassHunter Workstation software using m/z and retention time values obtained from the calibration samples to search for the targeted ions in the data. These searches generated extracted ion chromatograms (EICs) based on the list of target compounds. Peaks were integrated and compared to the calibration curves to calculate the concentration. Calibration curves were calculated from the calibration samples, prepared in the same oMM medium as all the samples, and curve fitting for each compound resulted in fits with $R^2$ values of 0.999. 4-morpholineethanesulfonic acid (MES), the buffer compound in the oMM medium with constant concentration and not utilized by yeast, was used as an internal standard (IS) for concentration normalization.

## Acknowledgements

We thank L Acosta-Sampson and A Gokhale for helpful discussions, J Dueber for xylose utilization pathway plasmids, Z Baer, J Kuchenreuthe and M Maurer for helps in anaerobic fermentation, and S Bauer and A Ibañez Zamora for help with analytical methods. This work was supported by funding from the Energy Biosciences Institute (JHDC, NLG and YSJ) and by a pre-doctoral fellowship from CNPq and CAPES through the program 'Ciência sem Fronteiras' (R E).

## Additional information

### Competing interests

XL: A patent application related to some of the work presented here has been filed on behalf of the Regents of the University of California. JHDC: A patent application related to some of the work presented here has been filed on behalf of the Regents of the University of California. The other authors declare that no competing interests exist.

### Funding

| Funder | Grant reference number | Author |
|---|---|---|
| University of California Berkeley | Energy Biosciences Institute | Xin Li, Vivian Yaci Yu, Yuping Lin, Kulika Chomvong, Raíssa Estrela, Annsea Park, Julie M Liang, Elizabeth A Znameroski, |

| Funder | Grant reference number | Author |
|---|---|---|
|  |  | Joanna Feehan, Soo Rin Kim, Yong-Su Jin, N Louise Glass, Jamie HD Cate |
| Conselho Nacional de Desenvolvimento Científico e Tecnológico |  | Raíssa Estrela |

The funders had no role in study design, data collection and interpretation, or the decision to submit the work for publication.

### Author contributions

XL, VYY, EAZ, JHDC, Conception and design, Acquisition of data, Analysis and interpretation of data, Drafting or revising the article; YL, KC, RE, AP, JML, JF, Acquisition of data, Analysis and interpretation of data, Drafting or revising the article; SRK, Analysis and interpretation of data, Drafting or revising the article, Contributed unpublished essential data or reagents; Y-SJ, Conception and design, Analysis and interpretation of data, Drafting or revising the article, Contributed unpublished essential data or reagents; NLG, Conception and design, Analysis and interpretation of data, Drafting or revising the article

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
