## [Decision Letter]

Thank you for sending your work entitled “Expanding xylose metabolism in yeast for plant cell wall conversion to biofuels” for consideration at *eLife*. Your article has been favorably evaluated by Detlef Weigel (Senior editor) and two reviewers, one of whom is a member of our Board of Reviewing Editors.

The Reviewing editor and the other reviewer discussed their comments before we reached this decision, and the Reviewing editor has assembled the following comments to help you prepare a revised submission.

We think your manuscript was significantly improved and has now given some fundamental insights into novel metabolic intermediates. However, there are several points that we ask you to consider for further improvement. Specifically, we would like to see that either the Thevelein strain is used (e.g. Biotechnol Biofuels, 2013, Jun 21; 6(1):89) and the experiments regarding the XI-based pathway is repeated, or that the entire paragraph about XI (paragraph ten of the Introduction, with Figure 5) is removed. In addition, please discuss the potential advantages of an XI-based pathway in connection with xylodextrin fermentation.

Minor comments:

Reviewer #1:

There are a couple of points, which I still recommend to reconsider:

a) In the Abstract, the finding of the xylosyl-xylitol intermediates in other organisms should be mentioned, underlining the broader finding of the manuscript.

b) The manuscript of Ryan et al. should be cited.

c) Figure 1—figure supplement 2: Growth of *N. crassa*. The figure is rather poor, the reduced growth *of N. crassa* Δcdt-2 on xylan is hardly visible. I suggest you provide a proper biomass measurement.

d) Figure 1—figure supplement 6: It is difficult to believe that the *N. crassa* GH43-3 is so different from the other enzymes that it appears more closely related to enzymes of basidiomycetes (*Schizophyllum*) than to other ascomycetes.

e) Figure 5—figure supplement 1: SDs of data are missing.

Reviewer #2:

When you state that “The excess reducing power generated by the XR/XDH pathway, initially deemed a problem…”, the XR/XDH pathway does not generate excess reducing power. The problem of the pathway is rather the cofactor imbalances between NADH and NADPH. This sentence should be omitted.

[Editors’ note: a previous version of this study was rejected after discussions between the reviewers, but the authors submitted for reconsideration. The previous decision letter is shown below.]

Thank you for choosing to send your work entitled “Expanding xylose metabolism in yeast for plant cell wall conversion to biofuels” for consideration at *eLife*. Your full submission has been evaluated by Detlef Weigel (Senior editor) and two peer reviewers, one of whom is a member of our Board of Reviewing Editors, and the decision was reached after discussions between the reviewers. We regret to inform you that your work will not be considered further for publication.

The reviewers had several concerns about your manuscript. Reviewer *#*1 indicated that some parts of the findings have already been published and the manuscript is not conceptually new. Also, the reviewer questioned the relevance of the novel oligosaccharides identified. Reviewer *#*2 was more positive. However, at the end also this reviewer questioned the break-through provided in your manuscript. In summary, we do not feel that the manuscript in its present version has sufficient novelty and quality for *eLife*. However, we could imagine being interested in a manuscript describing the combination of both pathways for xylose catabolism in a genetically engineered yeast, which should show a superior capacity to degrade xylose and to produce ethanol.

Reviewer #1:

In the current manuscript the authors describe the continuation of their work on the generation genetically engineered *Saccharomyces cerevisiae* (yeast) strains which are able to concert xylose to ethanol. The authors transferred a xylodextrin transporter, an intracellular β-xylosidase and an intracellular xylosyl-xylitol-specific ecxylosidase from the model fungus *Neurospra crassa*, which can grow on xylose, to *S. cerevisiae*. Production of the encoded proteins in an *S. cerevisiae* strain transformed with an XR/XDH/XK-pathway from *S. stipitis* resulted in the utilization of xylodextrins as the sole carbon source.

Major comments

1) Although I think in general genetic engineering of yeast to obtain strains with properties to degrade xylose is interesting, unfortunately, a very similar approach was published in 2010, by the same group, for the utilization of cellodextrin by recombinant *S. cerevisiae* by expressing a cellodextrin transporter and an intracellular β-glucosidase. Now, the authors applied the very same principle to the degradation of xylodextrin.

2) Furthermore, the here analyzed xylodextrin transporter CDT-2 was recently characterized in detail by [6] (PLOS One 9:e89330).

3) Also, the expression of 93xylosidase in yeast was reported before (Biosci Biotechnol Biochem, 2011, 75,1140-6). Thus, the only novelty of this manuscript is the production of xylosyl-xylitol oligomers as a side-activity of *S. stipitis* and *N. crassa* xylose reductases (XR). These oligomers appear to be inhibitory in engineered *S. cerevisiae* to utilize xylodextrins. The authors solved this problem by the identification and expression of a new hydrolyase, GH43-7, which can cleave the xylosyl-xylitol oligomers. Although the authors claim that these oligomers appear to be widely distributed in nature, evidence for this statement is lacking. By contrast, the production of these oligomers might just result from the strategy applied. In general, *S. cerevisiae* can be engineered to consume xylose either by expressing a xylose isomerase (XI) or by applying the XR/XDH pathway. Surprisingly, the authors used the XR/XDH pathway, although it is generally accepted that the XI pathway is more favorable for yeast (Biotechnol Biofuels, 2013, 6, 89; Metab Eng., 2012, 14: 611-22). As far as I know, the available xylose fermenting yeast strains from industry (DSM, Lesaffre, Terranol) are all based on the XI strategy. When the authors had used the XI strategy, they never would have encountered the problem of generating the xylosyl-xylitol oligomers by the heterologous XR.

Reviewer #2:

This paper describes expression of a novel pathway for metabolism of xylobiose and other short chain xylans in yeast. The pathway was identified in *N. crassa* and then transferred to yeast. The pathway involves two transporters and three intracellular hydrolysases, which were all identified in *N. crassa* based on RNAseq analysis. The xylose consumption pathway relies on the less efficient xylose reductase and xylitol dehydrogenase pathway, but it still seems to work due to the partly reduction of xylodextrins intracellularly. This activity is, however, only conferred by the *P. stipidis* XR, so the pathway for uptake of xylodextrins could work equally well (and probably better) with the xylose isomerase pathway, which is to be preferred as it does not result in accumulation of xylitol (also here observed by the authors, but not commented).

I think this paper is a fantastic new contribution to the field and it definitely serves consideration. Even though it could be interesting to combine the new pathway with the XI route for xylose catabolism, I do not think it is fair to ask the reviewers to demonstrate this. However, I suggest that they clearly state that the main benefit of their work is actually that they have found a way to overcome one of the most fundamental problems in xylose catabolism, namely an alternative transport of xylose into the cell. Xylose is transported through the HXT transporters and has competitive inhibition by glucose. This pathway may work in concert with glucose consumption, and this may be the real benefit. The data presented by the authors even indicate this. I suggest that this is clearly stated in the paper.

Besides this I do not have any major comments.

---

## [Author Response]

*We think your manuscript was significantly improved and has now given some fundamental insights into novel metabolic intermediates. However, there are several points that we ask you to consider for further improvement. Specifically, we would like to see that either the Thevelein strain is used (e.g. Biotechnol Biofuels, 2013, Jun 21; 6(1):89) and the experiments regarding the XI-based pathway is repeated, or that the entire paragraph about XI (paragraph ten of the Introduction, with*
Figure 5*) is removed. In addition, please discuss the potential advantages of an XI-based pathway in connection with xylodextrin fermentation*.

We thank the reviewers for their positive feedback on our manuscript. Although we agree it will be important to compare multiple XR/XDH and XI strains to gauge their performance, we think these experiments would fit better in a separate paper. We have therefore removed the paragraph and experiments regarding the XI-based pathway from our manuscript (paragraph beginning with “It has been proposed”, along with Figure 6, Figure 6–figure supplement 1, Figure 6–figure supplement 2, Table 1 and the associated Methods sections). We moved the observation of decreased xylitol production to the previous paragraph, where Figure 5—figure supplement 2 follows logically.

We also added a sentence and cited [43] on the potential use of the XI-based pathway in the penultimate paragraph of the main text.

*Minor comments*:

Reviewer #1:

*There are a couple of points, which I still recommend to reconsider*:

*a) In the Abstract, the finding of the xylosyl-xylitol intermediates in other organisms should be mentioned, underlining the broader finding of the manuscript*.

We thank the reviewer for this suggestion. We have changed the Abstract accordingly*.*

*b) The manuscript of Ryan et al. should be cited*.

We added this reference to the final paragraph, as it is indeed something worth pursuing for optimizing this pathway.

*c)*
Figure 1—figure supplement 2*: Growth of* N. crassa*. The figure is rather poor, the reduced growth of* N. crassa *Δcdt-2 on xylan is hardly visible. I suggest you provide a proper biomass measurement*.

We actually have the biomass data on xylan medium, but thought the photos would give the reader a better impression. We have now added the biomass data as panel B and worked to improve the image contrast of the original photos using linear scaling of the colors.

*d)*
Figure 1—figure supplement 6*: It is difficult to believe that* the N. crassa *GH43-3 is so different from the other enzymes that it appears more closely related to enzymes of basidiomycetes (*Schizophyllum) *than to other ascomycetes*.

This is an interesting point. GH43-3 is only 34% identical to the *Schizophyllum* protein over 85% of latter protein’s length. Using a simple PSI-BLAST search cannot detect *N. crassa* GH43-2 without significant phylogenetic support. Very likely, GH43-3 and GH43-2 are members of the same glycosyl hydrolase superfamily, but have very different functions.

*e)*
Figure 5—figure supplement 1*: SDs of data are missing.*

We carried out this experiment with three different xylose to xylodextrin ratios and saw the same pattern of consumption. This figure is a representative experiment, and we have indicated this in the figure legend.

Reviewer #2:

*When you state that “The excess reducing power generated by the XR/XDH pathway, initially deemed a problem…”, the XR/XDH pathway does not generate excess reducing power. The problem of the pathway is rather the cofactor imbalances between NADH and NADPH. This sentence should be omitted*.

We can see why the reviewer does not like this sentence, as it is not very precise. We have reworded the sentence to more accurately reflect what is going on, i.e. that cofactor imbalance leads to accumulation of reduced byproducts (xylitol and glycerol). We do think it’s worth keeping this sentence in the paper, as realistic hydrolysates will likely have a mixture of xylose and xylodextrins, and we have conclusively demonstrated that the cofactor imbalance of XR/XDH pathway can be exploited to drive reduction of acetate to ethanol (45).

[Editors’ note: the author responses to the previous round of peer review follow.]

In the last round of review, the reviewers were not convinced that the xylodextrin consumption pathway we identified in *N. crassa* and the xylosyl-xylitol metabolic intermediates were widespread in nature. Furthermore, the reviewers were not convinced that this pathway could be competitive with one using xylose isomerase (XI) as opposed to xylose reductase plus xylitol dehydrogenase (XR/XDH). We have now carried out a large number of experiments to address these concerns.

First, we tested whether xylosyl-xylitols are generated intracellularly in a number of fungi, as well as in *Bacillus subtilis*. As shown in the new Figure 3 and Figure 3—figure supplement 1, we find xylosyl-xylitols present in all of these organisms when they are grown on xylodextrins. These experiments convincingly show that xylosyl-xylitols are in fact widespread in nature, generated by microbes spanning well over 1 billion years of evolution. They are not simply a byproduct of reconstituting xylodextrin consumption in *S. cerevisiae*.

Second, we carried out a direct comparison between two of the best xylose fermentation strains in the public domain–strain SR8, which uses the XR/XDH pathway (see [24], PLOS One), and strain SXA-R2P-E which uses the XI pathway and which was kindly provided by Prof. Hal Alper (http://www.biotechnologyforbiofuels.com/content/7/1/122). Using anaerobic conditions, we find that, regardless of starting conditions, cell loading, or media, strain SR8 is far superior to strain SXA-R2P-E in terms of ethanol productivity. Strain SXA-R2P-E has a slightly higher yield of ethanol and produces less xylitol. To our knowledge, this is the first direct experimental comparison between high-performing XI and XR/XDH strains in the literature, and will be of wide interest.

Remarkably, we were surprised to find that, when using xylodextrins, the XR/XDH pathway produces far less xylitol when compared to xylose fermentations (Figure 5 and Figure 5—figure supplement 2). Thus, not only does the capacity of the XR/XDH pathway for high flux far exceed that of the XI pathway, the propensity of the XR/XDH pathway to produce xylitol can be reduced by using xylodextrins as opposed to xylose in fermentations. Thus, we think it is highly worthwhile to explore the potential of the xylodextrin pathway we’ve identified for the first time. Although the flux through the upstream part of the xylodextrin consumption pathway is not as high as with xylose fermentation, we think future efforts such as directed evolution of the xylodextrin transporter can be used to solve this problem (see Ryan, 2014, eLife 3:e03703).

Finally, we now show that the xylodextrin pathway can be exploited in a range of cofermentation scenarios, including glucose plus xylodextrins (suggested by reviewer *#*2) and sucrose plus xylodextrins. These are shown in Figure 5 and Figure 7. Cofermentation of xylose equivalents with glucose equivalents is a “holy grail” in the biofuel field, and we have now at least doubled the options available for future exploration (two previously described scenarios include glucose plus xylose and cellobiose plus xylose).

In summary, for both the ecological insights regarding xylodextrin consumption by microbes in nature, and the possible applications of xylodextrin fermentation to produce biofuels, we think this paper should be published in *eLife*.

Reviewer #1:

*In the current manuscript the authors describe the continuation of their work on the generation genetically engineered* Saccharomyces cerevisiae *(yeast) strains which are able to concert xylose to ethanol. The authors transferred a xylodextrin transporter, an intracellular β-xylosidase and an intracellular xylosyl-xylitol-specific β-xylosidase from the model fungus* Neurospra crassa*, which can grow on xylose, to* S. cerevisiae*. Production of the encoded proteins in an* S. cerevisiae *strain transformed with an XR/XDH/XK-pathway from* S. stipitis *resulted in the utilization of xylodextrins as the sole carbon source.*

Major comments

*1) Although I think in general genetic engineering of yeast to obtain strains with properties to degrade xylose is interesting, unfortunately, a very similar approach was published in 2010, by the same group, for the utilization of cellodextrin by recombinant* S. cerevisiae *by expressing a cellodextrin transporter and an intracellular β-glucosidase. Now, the authors applied the very same principle to the degradation of xylodextrin*.

We agree that the approach we have used here is somewhat similar to that used in 2010 for the utilization of cellobdextrins. However, we note that this is the approach evolved in *N. crassa* for optimal growth on the plant cell wall, and involves an entirely surprising metabolic intermediate that had not been identified before. Furthermore, we now show this to be a very general pathway for consuming xylose. It is used in fungi spanning evolutionary time to the likely advent of the general architecture of the plant cell wall now extant in plants. Furthermore, the pathway is even present in bacteria, as we have now shown using *B. subtilis* and phylogenetic analysis of other bacteria (see Figure 3 and Figure 3—figure supplement 1). Thus, this pathway has arisen in organisms that span well over a billion years of evolutionary history. From a fundamental science perspective, we think this is certainly conceptually new and worth publishing in *eLife*.

*2) Furthermore, the here analyzed xylodextrin transporter CDT-2 was recently characterized in detail by*
[6]
*(PLOS One 9:e89330)*.

As noted above, xylodextrin transport is only a small part of the story. Furthermore, without identifying the xylosyl-xylitol intermediates, the PLOS ONE paper is far from complete, and misses a major aspect of how microbes grow on plants. Again, the discovery that organisms as diverse as *B. subtilis* and many fungi produce these intermediates is entirely new, and was missed by every one who has worked on xylose metabolism.

*3) Also, the expression of β-xylosidase in yeast was reported before (Biosci Biotechnol Biochem, 2011, 75,1140-6). Thus, the only novelty of this manuscript is the production of xylosyl-xylitol oligomers as a side-activity of* S. stipitis *and* N. crassa *xylose reductases (XR). These oligomers appear to be inhibitory in engineered* S. cerevisiae *to utilize xylodextrins. The authors solved this problem by the identification and expression of a new hydrolyase, GH43-7, which can cleave the xylosyl-xylitol oligomers. Although the authors claim that these oligomers appear to be widely distributed in nature, evidence for this statement is lacking. By contrast, the production of these oligomers might just result from the strategy applied*.

As we have now shown, xylosyl-xylitol oligomers are generated by a wide variety of microbes spanning over a billion years of evolution. We think this is a fundamental discovery worth publishing in *eLife*.

*In general,* S. cerevisiae *can be engineered to consume xylose either by expressing a xylose isomerase (XI) or by applying the XR/XDH pathway. Surprisingly, the authors used the XR/XDH pathway, although it is generally accepted that the XI pathway is more favorable for yeast (Biotechnol Biofuels, 2013, 6, 89; Metab Eng., 2012, 14: 611-22). As far as I know, the available xylose fermenting yeast strains from industry (DSM, Lesaffre, Terranol) are all based on the XI strategy. When the authors had used the XI strategy, they never would have encountered the problem of generating the xylosyl-xylitol oligomers by the heterologous XR*.

We strongly disagree that the XI pathway has been shown to be superior to the XR/XDH pathway. There have been few if any direct comparisons of these pathways in the literature, with what could be described as state-of-the-art strains. To overcome this deficiency we have used two of the best strains available in the public domain in head-to-head comparisons. For the XR/XDH pathway, we used strain SR8, which is one of the best reported strains for xylose fermentation. For the XI pathway, we used strain SXA-R2P-E from Prof. Hal Alper’s lab, which he kindly provided us for these comparisons. This strain was recently reported in Biotechnology for Biofuels, and is one of the two best XI strains in the public domain (the one from Metab. Eng., 2012, 14: 611-22 is not available, although we requested it).

We used a number of conditions, all anaerobic, to compare the two strains head-to-head in batch fermentations. We find that, regardless of starting OD (or g cells/L culture), the SR8 strain expressing the XR/XDH pathway is far superior to the XI strain, with roughly twice the ethanol productivity. By contrast the XI strain is slightly better in terms of ethanol yield for the XR/XDH strain (see Figure 6, Figure 6–figure supplement 1, Figure 6–figure supplement 2 and Table 1 in the new manuscript). Thus, even prior to considering the use of xylodextrins, the XR/XDH pathway is highly competitive, if not outright superior, to the XI pathway. Thus the rationale put forward by reviewer #1 that the XI pathway is preferred is not obvious.

We have also made the surprising observation that using xylodextrins minimizes xylitol byproduct formation by the XR/XDH pathway, which was previously seen as a fundamental weakness of the pathway. See Figure 5 and Figure 5—figure supplement 2. This suggests that, with further optimization of the upstream part of the pathway, i.e. through directed evolution of the transporter (see our recent work in eLife, Ryan et al.), the XR/XDH pathway could achieve both high ethanol productivity and yield, eliminating the rationale for the use of XI.

In addition, we discussed with Dr. Amit Gokhale from BP scenarios that could be of immediate use in improving biofuel production. He noted that, as we indicated in our original manuscript, xylodextrin fermentation could be used as an add-on to existing sugarcane ethanol production processes. In short, hot water pretreatment could be used to release xylodextrins from bagasse, and the resulting water could be used as the diluent for the cane juice used in ethanol fermentations. This would have the benefit of increasing ethanol yield beyond that possible from using cane juice alone. Indeed, we find this to be the case (see Figure 7 in the new manuscript). Thus, cofermentation of sucrose and xylodextrins is a new and promising direction for producing biofuels.

Reviewer #2:

*This paper describes expression of a novel pathway for metabolism of xylobiose and other short chain xylans in yeast. The pathway was identified in* N. crassa *and then transferred to yeast. The pathway involves two transporters and three intracellular hydrolysases, which were all identified in* N. crassa *based on RNAseq analysis. The xylose consumption pathway relies on the less efficient xylose reductase and xylitol dehydrogenase pathway, but it still seems to work due to the partly reduction of xylodextrins intracellularly. This activity is, however, only conferred by the* P. stipidis *XR, so the pathway for uptake of xylodextrins could work equally well (and probably better) with the xylose isomerase pathway, which is to be preferred as it does not result in accumulation of xylitol (also here observed by the authors, but not commented)*.

As described above in response to reviewer #1, the XI pathway is inferior to the XR/XDH pathway with respect to productivity, by a wide margin. We also note that, to our surprise, the XR/XDH pathway does not generate nearly as much xylitol when xylodextrins are used as a carbon source rather than xylose. This could in fact reflect one of the key advantages of using xylodextrins in preference to xylose, both in nature and in an industrial setting.

*I think this paper is a fantastic new contribution to the field and it definitely serves consideration. Even though it could be interesting to combine the new pathway with the XI route for xylose catabolism, I do not think it is fair to ask the reviewers to demonstrate this. However, I suggest that they clearly state that the main benefit of their work is actually that they have found a way to overcome one of the most fundamental problems in xylose catabolism, namely an alternative transport of xylose into the cell. Xylose is transported through the HXT transporters and has competitive inhibition by glucose. This pathway may work in concert with glucose consumption, and this may be the real benefit. The data presented by the authors even indicate this. I suggest that this is clearly stated in the paper*.

We thank the reviewer for the positive feedback. As we describe above in response to reviewer #1, we have directly compared XR/XDH and XI pathways and found the XR/XDH pathway to be superior in terms of productivity. We also show that the use of xylodextrins reduces the amount of xylitol byproduct produced by the strain expressing XR/XDH. As suggested by the reviewer, we have now shown that glucose plus xylodextrin cofermentation is indeed possible, and produces very little xylitol as a byproduct (Figure 5). We have also used xylodextrins in combination with sucrose, and see increased ethanol yields that are very promising in terms of an application to enhance sugarcane ethanol production (Figure 7).